# A Spoken Language Dataset of Descriptions for Speech-Based Grounded Language Learning

**Gaoussou Youssouf Kebe,**[1]* **Padraig Higgins,**[1]* **Patrick Jenkins,**[1]* **Kasra Darvish,**[1] **Rishabh Sachdeva,**[1] **Ryan Barron,**[1] **John Winder,**[1,2] **Don Engel,**[1] **Edward Raff,**[1,3] **Francis Ferraro,**[1] **Cynthia Matuszek**[1]

[1]University of Maryland, Baltimore County
[2]Johns Hopkins Applied Physics Laboratory
[3]Booz Allen Hamilton

## Abstract

Grounded language acquisition is a major area of research combining aspects of natural language processing, computer vision, and signal processing, compounded by domain issues requiring sample efficiency and other deployment constraints. In this work, we present a multimodal dataset of RGB+depth objects with spoken as well as textual descriptions. We analyze the differences between the two types of descriptive language and our experiments demonstrate that the different modalities affect learning. This will enable researchers studying the intersection of robotics, NLP, and HCI to better investigate how the multiple modalities of image, depth, text, speech, and transcription interact, as well as how differences in the vernacular of these modalities impact results.

## 1 Introduction

Grounded language acquisition is the process of learning language as it relates to the world—how concepts in language refer to objects, tasks, and environments [46]. *Embodied* language learning specifically is a significant field of research in NLP, machine learning, and robotics. While there is substantial current effort on learning grounded language for embodied agents [11, 28, 63], in this work we describe learning from multiple modalities, including text, transcribed speech, and speech audio.

Text is a common input domain when learning grounded language, yet many systems use speech once deployed [75]. In practice, embodied agents are likely to need to operate on imperfectly understood spoken language. Speech-based assistive devices have gained significant popularity in the last few years, representing perhaps the first widely deployed, communicative 'embodied agents' in human environments. Spoken language is critical for interactions in physical contexts, despite the inherent difficulties: spoken sentences tend to be less well framed than written text, with more disfluencies and grammatical flaws [56].

There are many ways in which robots learn grounded language [12, 16, 30, 43, 73, 76, 80], but they all require either multimodal data or natural language data—usually both. Current approaches to grounded language learning require data in both the perceptual ("grounded") and linguistic domains. While existing datasets have been used for this purpose [16, 31, 33, 51, 74], the language component is almost always derived from either textual input or manually transcribed speech [44, 73].

To that end, we present the **Gro**unded **L**anguage **D**ataset (GOLD), which contains images of common household objects and their description in multiple formats: text, speech (audio), and speech transcriptions (see fig. 1). The primary contributions of this paper are as follows:

---

*Equal contributions from first three authors

35th Conference on Neural Information Processing Systems (NeurIPS 2021) Track on Datasets and Benchmarks.

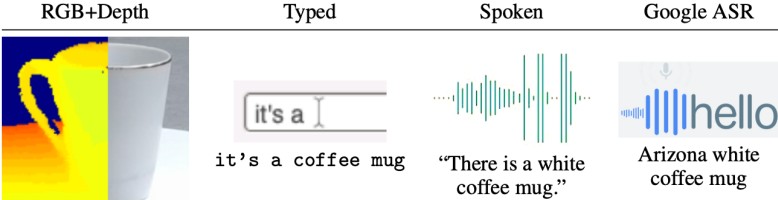

| RGB+Depth | Typed | Spoken | Google ASR |
|---|---|---|---|
| | it's a | | hello |
| | it's a coffee mug | "There is a white coffee mug." | Arizona white coffee mug |

Figure 1: GOLD has RGB and depth point cloud images of 207 objects in 47 categories. It includes 16500 text and 16500 speech descriptions; all spoken descriptions include automatic transcriptions.

1. We provide a publicly available, multimodal, multi-labelled dataset of household objects, with image+depth data and textual and spoken descriptions with automated transcription.

2. We show that learning language groundings from transcribed or raw speech performs similarly to models trained on typed text, while allowing those descriptions to be provided in a more natural, convenient way.

3. We demonstrate that the dataset poses a number of interesting research challenges including identifying bias in speech processing from the unique perspective of language grounding.

## 2 Related Work

Language acquisition can be used for interactions with robots [2, 10, 43, 73]. On a robot, the grounded language acquisition task has a number of uses. Retrieving objects based on their descriptions [52] is a necessary component of caretaking and domestic robots. Grounding landmarks and instructions can aid robots in navigation of novel spaces [73, 79]. [71] surveyed the many machine learning methods used, possible applications, and the human-robotic interaction implications of grounded language learning on a robotic platform.

Grounded language acquisition is in the unique position of requiring a dataset that combines sensory perception with language. These combined datasets are frequently handcrafted for the specific task that the research seeks to accomplish [12, 59], often leading to narrower applications. For example, CLEVR [31] was designed as a benchmark for question answering tasks. Objects in CLEVR are limited to a small set of attributes which in turn limits the types of questions in both syntax and content. In comparison, GOLD contains more complex real-world objects and does not limit the scope of the annotations to a fixed set of characteristics.

We note that the image component of GOLD is heavily influenced by the University of Washington RGB-D dataset [35]. Both datasets contain large numbers of everyday objects from multiple angles. Our dataset is collected with a now state of the art sensor which enables us to capture smaller objects at a finer level of detail (such as an Allen key, the diameter of which pushes the limits of the depth sensor when laid on the table). Additionally, we select objects based on their potential utility for specific human-robot interaction scenarios, such as things a person might find in a medicine cabinet or first aid kit, enabling learning research relevant to eldercare and emergency situations [8].

Creating a dataset that includes speech has a high cost of collecting and transcribing audio. [59] presents a grounded language system that can generate descriptions for targets within a scene of colored rectangles. The visual data for this task is easily generated, but speech descriptions were recorded and transcribed over a long period of time. The manual audio transcription task can take four to ten hours per hour of audio [21, 85]. Such perfectly transcribed audio is also unrealistic for real-world usage, which must rely on automation. We acknowledge this challenge, and we evaluate automatically-produced transcriptions for their quality. We also include the automatically-produced transcriptions along with the raw audio.

Recent datasets that include speech such as Flickr Audio Captions [24], SpokenCOCO [29], SPEECH-COCO [26], Synthetically Spoken COCO, Synthetically Spoken STAIR get around this by generating spoken descriptions from the text captions provided by the Flickr8K, COCO [38], and STAIR [86] datasets. Speech COCO, Synthetically Spoken COCO [15], and Synthetically Spoken STAIR [27] generate their speech through text to speech systems while Flickr Audio Captions and SpokenCOCO use crowdsourced workers. Places Audio Captions [25] which uses the MIT Places 205 Database [87]

Table 1: Classes of objects in GoLD.

| Topic | Classes of Objects |
|---|---|
| food | *potato, soda bottle, water bottle, apple, banana, bell pepper, food can, food jar, lemon, lime, onion* |
| home | *book, can opener, eye glasses, fork, shampoo, sponge, spoon, toothbrush, toothpaste, bowl, cap, cell phone, coffee mug, hand towel, tissue box, plate* |
| medical | *band aid, gauze, medicine bottle, pill cutter, prescription medicine bottle, syringe* |
| office | *mouse, pencil, picture frame, scissors, stapler, marker, notebook* |
| tool | *Allen wrench, hammer, measuring tape, pliers, screwdriver, lightbulb* |

is the only other dataset in this area where the speech is collected directly from the spoken descriptions of crowd workers, however the descriptions are of all the salient objects in an image instead of a single object. All these datasets also only contain color images while GoLD extends this to include depth images and pointclouds.

In our work we adopt the manifold alignment model form [49] which is similar to [52]. The latter trained a grounded language model in order to retrieve objects with a robotic arm from natural language descriptions. The robot learned the functionality of objects through text data gathered from Wikipedia.

Embodied approaches [2, 72] are important for collecting multimodal data on robotic platforms. [77] created a robot that learned from both language and sensed traits including the visual, proprioceptive, and auditory characteristics of objects. However, the language was used only to identify named objects. [4] developed a robot that memorized which objects it had seen before by combining multimodal data about the object including visual, haptic, and researcher provided linguistic percepts.

Recently, manifold alignment has been used and outperformed traditional classification methods, particularly for grounded language tasks [9, 49, 52]. One particular benefit of manifold alignment is that it enables arbitrary embeddings to be used and aligned. In contrast to prior grounding approaches, these embeddings do not have to be restricted to individual words, and instead can be computed for an entire input (e.g., utterance). As a result, we use *grounded language manifold alignment* techniques to experimentally validate GoLD.

## 3   GoLD: The Grounded Language Dataset

GoLD is a collection of visual and English natural language data in five high-level groupings: *food*, *home*, *medical*, *office*, and *tools*. In these groups, 47 object classes (see table 1) contain 207 individual object instances. The dataset contains vision and depth images of each object from 450 different rotational views. From these, four representative 'keyframe' images were selected. These representative images were used to collect 16500 textual and 16500 spoken descriptions. The dataset contents are summarized in table 2.

Table 2: Components of GoLD.

| Categories (*e.g.*, medicine) | 5 | Images (vision + depth) | 825 |
|---|---|---|---|
| Classes (*e.g.*, apple) | 47 | Text descriptions | 16500 |
| Object instances (*e.g.*, `apple_3`) | 207 | Spoken descriptions | 16500 |

Visual inputs were collected by rotating objects on a turntable in front of a commodity RGB-D (RGB + depth) video camera, as in [35]). For each object, four keyframes were manually selected to capture representative, diverse view angles of each object. Amazon Mechanical Turk workers were shown all four images and asked to provide either spoken or typed descriptions.

### 3.1   Vision + Depth Data Collection

Visual perception data were collected using a Microsoft Azure Kinect (i.e.., a Kinect 3), a low-cost, high-fidelity commodity sensor that is widely used in robotics. For each object instance (*i.e.*, for each of the four staplers in the dataset), this sensor was used to collect raw image and point cloud videos.

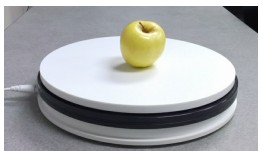 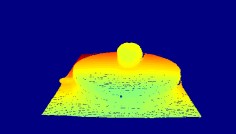 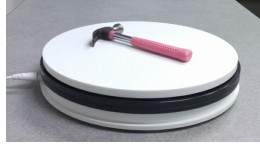 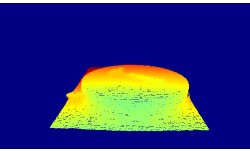

(a) Apple image frame.     (b) Apple point cloud.     (c) Hammer image frame.     (d) Hammer point cloud.

Figure 2: Samples showing keyframes in GOLD, along with the aligned 3D point cloud with depth information. Only the RGB image was shown to labelers.

The resulting dataset contains 207 90-second depth videos, one per instance, showing the object performing one complete rotation on a turntable. To ensure that each object has diverse views, e.g., examples of a mug with the handle occluded and visible, we manually selected 825 pairs of image and depth point cloud from 207 objects as representative frames, which we refer to as keyframes (examples are shown in fig. 2).

Manually selecting keyframes avoids a known problem with many visual datasets: their tendency to show pictures of objects taken from a limited set of 'typical' angles [7]. For example, it is rare for a picture of a banana to be taken end-on. This aligns with our motivation of creating a dataset of household objects to support research on grounded language learning in an unstaged environment, as a robot looking at an object in a home may not see this typical view.

### 3.2 Text and Speech Description Collection

All descriptions were collected using Amazon Mechanical Turk (AMT).[2] Keyframes for randomly-chosen object instances were shown to the worker. They were asked to either type descriptions of objects in one or two short, complete sentences, or record descriptions using a microphone.

Collected speech was transcribed using Google's Speech to Text API, resulting in a spoken-language corpus of 16500 verbal descriptions. It should be noted that, although Mechanical Turk does not provide personally identifiable information about workers, it is possible that users may be identified by their voice or other side-channel information. For this reason, all collected language is limited to factual descriptions of simple household objects, and no value judgments, opinions, or emotional or potentially damaging subjects are discussed.

#### 3.2.1 Speaker Voice Qualities

We collected spoken descriptions from 552 Amazon Mechanical Turk workers. We labeled each of these workers based on perceived gender (man, woman, or undetermined),[3] accent (whether the speaker has a non-mid-American accent), creak (whether the user has a raspy, low-register voice), hoarseness (whether the

| Quality | Value | Count |
|---|---|---|
| Perceived Gender | Men | 271 |
| | Women | 274 |
| | Undet. | 7 |
| Accent | Yes | 279 |
| | No | 273 |
| Creak | Yes | 194 |
| | No | 358 |
| Hoarseness | Yes | 48 |
| | No | 504 |
| Muffledness | 1 | 393 |
| | 2 | 119 |
| | 3 | 40 |
| Volume | 1 | 10 |
| | 2 | 157 |
| | 3 | 331 |
| | 4 | 54 |
| Background Noise | 1 | 366 |
| | 2 | 143 |
| | 3 | 39 |
| | 4 | 4 |

Table 3: Number of workers labeled with each characteristic.

speaker has a strained, breathy voice), muffled-ness (the level of distortion of the user's microphone, 1 to 3), volume (1 to 4), and level of background noise (1 to 4). Section 3.2 shows the number of workers to whom each label has been attributed.

We intend for this data to be used as a test-bed for bias studies and other research into the performance of grounding models for different sub-populations. For example, a pilot study on this data has shown that accented users are particularly affected by the bias of speech-to-text models and that learning directly from raw speech can mitigate this bias.

---

[2]See Ethical Considerations section, appendix.

[3]Gender and sex are complex constructs. We asked annotators to choose the category that seemed to 'best describe' the speaker, but acknowledge the limitations of this approach.

### 3.2.2 Accuracy of Speech Transcriptions

Obtaining accurate transcriptions of speech in sometimes noisy environments is a significant obstacle to speech-based interfaces [37]. To understand the degree to which learning is affected by ASR errors, 250 randomly selected transcriptions were manually evaluated on a 4-point scale (see table 4). Of those, 80% are high quality ('perfect' or 'pretty good'), while only 11% are rated 'unusable.'

To get a more detailed understanding of transcription accuracy, we compare the ASR transcriptions and the human-provided transcriptions using the standard word error rate (WER) [55] and Bilingual Evaluation Understudy (BLEU) [54] scores. BLEU scores are widely used to measure the accuracy of language translations based on string similarity; we adopt this system to evaluate the goodness of transcriptions. BLEU is calculated by finding $n$-gram overlaps between machine translation and reference translations. We use tri-grams for our BLEU scores since some descriptions are shorter than four words such as "these are pliers", rendering a 4-gram BLEU score meaningless.

Table 4: Human ratings of 250 automatic transcriptions. These ratings are designed strictly to assess the accuracy of the transcription, not the correctness of the spoken description with respect to the described object.

| Rating | Transcription Quality Guidelines | # |
|--------|----------------------------------|-----|
| 1 | wrong or gibberish / unusable sound file | 28 |
| 2 | slightly wrong (missing keywords / concepts) | 21 |
| 3 | pretty good (main object correctly defined) | 33 |
| 4 | perfect (accurate transcription and no errors) | 168 |

Figure 3 shows that many of the 250 manually transcribed descriptions were perfectly transcribed by automated speech-to-text. The marginal BLEU histogram shows more mistaken transcriptions (the second peak around 0) due to known problems with using BLEU to evaluate short sentences and tokens having mismatched capitalization or punctuation.

### 3.3 Comparative Analysis

Our initial hypothesis was that people would use more words when describing objects verbally than when typing, as it is lower effort to talk than to type. Accordingly, We find spoken descriptions to be slightly longer than their textual counterparts ($p \geq 13.71$ using a Welch's t-test) While speech has more average words per description, 11.7, compared to text at 10.46, when stop words are removed the averages are 6.1 and 5.89 respectively. The larger mean drop in the speech descriptions is likely due to the tendency of ASR systems to interpret noise or murmur utterances as filler words, the inclusion of which has been shown to detract from meaning [68]. Text descriptions are a more consistent length than speech, with a standard deviation of 6.7 words for

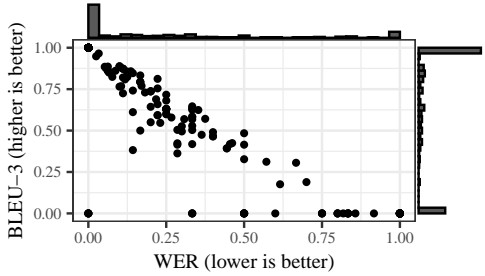

Figure 3: BLEU-3 and WER scores for 250 randomly selected speech transcriptions. A WER of 0 and a BLEU of 1 (top left corner) indicates perfect transcription. Marginal histograms show that some descriptions were perfectly transcribed.

text, versus 9.51 for transcribed speech. When we remove stop words, the standard deviation is 3.63 for text and 4.69 for speech.

Table 5 shows the top 20 most frequent words in both modalities. There is substantial overlap, as expected, since the same objects are being described, with colors dominating the lists. People use more filler words when describing the objects using speech; for example, the word 'like' appears 889 times in speech data whereas it was not significant in the text data.

Using the Stanford Part-of-Speech Tagger [78] to count the number of nouns, adjectives, and verbs between the two modalities yields no significant differences between the modalities. However, the word 'used' appears frequently (see table 5), typically to describe functionality. Developing grounded language models around functionality for the analysis of affordances in objects [52] is an important research avenue that our dataset enables, which is not possible in prior datasets that do not contain the requisite modalities.

### 3.4 Dataset Distribution

The data is publicly available as a GitHub repository[4]. The repository contains three high-level datatypes: perception and language. The perceptual data is split into RGB-D images and depth data in the form of point clouds [60]. Each of these sets of data is subdivided by object class (e.g., "apple") and then further by instance (e.g., "apple #4"). The language is subdivided similarly, and for each object instance contains multiple speech descriptions (as .wav files) along with ASR transcriptions of that speech. Each instance also has multiple associated typed descriptions, which are not related to the spoken descriptions—they were provided by different workers at a separate time.

Each description of an instance also includes associated meta-data describing the data collection process. This includes: (1) a numeric identifier for the worker who provided each description; (2) the amount of time each description took to provide; and (3) the ground-truth category and instance label for each object.

Table 5: Top 20 most frequently used words in text (left) and speech (right) by percentage of occurrence in descriptions.

| Token | % Frequency | Token | % Frequency |
|---|---|---|---|
| black | 13.24 | black | 13.92 |
| white | 10.66 | white | 12.85 |
| blue | 9.97 | blue | 10.23 |
| bottle | 9.50 | red | 9.13 |
| red | 9.45 | yellow | 8.97 |
| yellow | 9.02 | bottle | 8.50 |
| object | 7.99 | small | 7.96 |
| small | 6.44 | used | 7.21 |
| green | 5.82 | object | 6.41 |
| pair | 5.27 | green | 5.85 |
| used | 5.21 | plastic | 5.30 |
| handle | 4.58 | color | 5.22 |
| plastic | 4.40 | handle | 4.85 |
| silver | 3.88 | like | 4.62 |
| box | 3.69 | looks | 3.99 |
| label | 2.92 | silver | 3.66 |
| metal | 2.79 | turntable | 3.33 |
| pink | 2.66 | pair | 3.32 |
| light | 2.44 | box | 3.21 |
| scissors | 2.43 | label | 3.01 |

## 4   Experiments

GOLD is designed to enable multiple research directions. In our evaluation we will demonstrate initial baseline results for classification, retrieval, and speech recognition tasks that are enabled by GOLD. Each experiment will combine the RGB+depth images with one of the three language domains: text, transcribed speech, and speech audio. We also perform a fourth learning experiment on a combination of text and transcribed speech to test how the combination of the two might boost learning. In each case the goal is to learn how to ground the unconstrained natural language descriptions of objects with their associated visual percepts of color and depth. This allows research investigating the impact of information lost via reductions from raw speech, to text, to noisier ASR text. The textual inputs naturally lack the inflection and tonal characteristics that will be critical for user interaction with a robot, but not easily studied with current datasets. Since speech is a natural mode of communication for humans, and information such as inflection are lost after transcription, we would like to move in a direction where speech audio is the primary input into our models, forgoing transcription entirely.

**Manifold Alignment.** As noted in section 2, we use manifold alignment [3, 82, 83] with triplet loss [6, 49] to embed the visual percepts and language data from GOLD into a shared lower dimensional space. Within this space, a distance metric is applied to embedded feature vectors in order to tell how well a particular utterance describes an image, with shorter distances implying a better description. The manifold alignment model is shown in fig. 4.

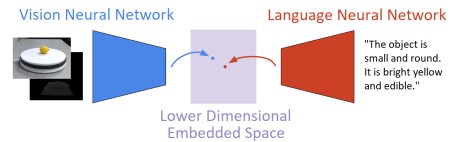

Figure 4: A high-level view of a manifold alignment model. Vision and language domains are embedded into a shared lower dimensional space. Pairs of vision and text are aligned to be closer to each other within the embedded space.

For example, a picture of a lemon and the description "The object is small and round. It is bright yellow and edible." should be closer together in the embedded space than the same picture of a lemon and the unrelated description "This tool is used to drive nails into wood," since the latter description was used to describe a hammer. Through this technique, even novel vision or language inputs should be aligned, meaning that a new or different description of a lemon should still be closely aligned in the embedded space. We would additionally expect other similar objects, such as an orange, to be described in a somewhat similar way, allowing for potential future learning of categorical information.

---

[4]https://github.com/iral-lab/gold

**Vision.** The vision feature vectors are created following the work of [20] and [58]. Color and depth images are passed through CNNs that have been pretrained on ImageNet [17] with the prediction layer removed so that the final layer is a learned feature vector. Depth images are "colorized" to enable classification re-using the same network [58]. The two vectors, one from color and one from depth, are concatenated into a 4096-dimensional visual feature vector.

**Text and Speech Transcriptions.** The language features of text and transcribed speech data are extracted using BERT [18]. Each natural language description is fed to a BERT pretrained model. We obtain the final embedding by concatenating the last four hidden layers of BERT. The resulting 3072-dimensional vector is taken as the description's language feature vector and associated to the visual feature vector of the frame it describes.

**Speech.** Self supervised pretrained models inspired by NLP methods have recently shown success in speech representation. We use wav2vec 2.0 [5], a self-supervised speech model that learns over continuous representations of raw speech through a BERT [18] inspired masked language modeling task. Similarly to the text featurization, features are then learned by performing average-pooling over the concatenation of the last four layers of the transformer.

To evaluate the benefit of using a pre-trained model, we also consider 40 dimensional Mel-frequency cepstral coefficient (MFCC) features [47] that are extracted from the raw audio with a 10 ms frame shift. Due to the lower-dimensional nature of MFCCs, the language network is modified to include a Long Short-Term Memory (LSTM) network. 64-dimensional outputs from the final 32 hidden states [13] are concatenated together to form a fixed length 2048-dimensional speech vector which are passed to a fully connected layer and output into the same embedded dimension as the visual network.

**Triplet Loss.** The triplet loss function [6, 62] uses one training example as an "anchor" and two more points, one of which is in the same class as the anchor (the positive), and one which is not (the negative). For example, while classifying tools the anchor might be a hammer, the positive would be a different hammer, and the negative would be a screwdriver. The loss function then encourages the network to align the anchor and positive in the embedded space while repelling the anchor and the negative. In order to align the networks to each other and keep each network internally consistent, the anchor, positive, and negative instances are chosen randomly (balanced across cases) from either the vision or language domains at training time.

For anchors ($A$), positive instances ($P$), and negative instances ($N$), we compute embeddings of these points, then compute triplet loss in the standard fashion with a default margin $\alpha = 0.4$ [62] where $f$ is the relevant model for the domain of the input:

$$\mathcal{L} = \max(0, \|f(A) - f(P)\|^2 - \|f(A) - f(N)\|^2 + \alpha) \tag{1}$$

**Training.** Five models are trained from the data. Each combines the visual data with a different language domain out of text, transcribed speech, text + transcribed speech, and speech audio. Vision data are matched with language data by their instance names and approximately 80% is reserved for training, 10% for validation and 10% for testing for a total of 13,040 text and speech training examples.

The models are trained with the ADAM optimizer on a 20GB Quadro RTX 6000 GPU. Each model is trained for a different number of epochs to balance for the variation in size of the training sets. Text, transcribed speech and speech are trained for 300, and the combined text and transcribed speech model is trained for 150 epochs. Each model outputs into a 1024-dimensional embedded space.

## 5   Evaluation

A held out testing set containing at least one of each object class is used for evaluation. A given image can only appear in one of the training and testing sets. We have found that the manifold alignment approach does not perform well on unseen object classes. We evaluate the models in two ways. We calculate the precision, recall, and F1 metrics by classifying based on the proximity to a target embedded datum. This method is further explained in Section 5.1. Finally, we calculate the Mean Reciprocal Rank (MRR) of two mock object retrieval tasks.

## 5.1 Grounded Language as Classification

The manifold alignment models we employ from [49] do not output a binary yes/no classification. Instead, classification is based on the proximity within the embedded space. This raises the question of how to define when two embedded vectors are "close" enough to be classified as related. To test this, we normalize the distances within our validation set to be between 0 and 1 by dividing the cosine distance by 2. Given a distance threshold between 0 and 1, we then classify positive instances as being within the threshold distance and negative instances as being outside the threshold. We then calculate the precision, recall, and F1 measure on our testing data as a function of the threshold. The F1 score at different thresholds for the combined text and speech transcription model can be seen in section 5.1.

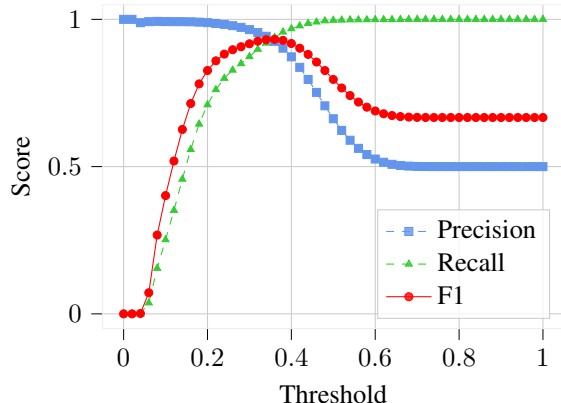

Figure 5: Precision, recall, and F1 on the validation set as a function of the threshold for classification for a combination of text and transcribed speech (peak F1=0.93). Graphs for pure text and pure speech show a very similar shape, reaching peak F1 of 0.91 and 0.94 respectively.

We see the best F1 results on the validation set with thresholds in the range $[0.35, 0.45]$. When those thresholds are applied to the testing set, the F1 for the text, transcribed speech, and combined models are .84, .94, and .92, respectively as shown in Table 6.

## 5.2 Grounded Language as Retrieval

The Mean Reciprocal Rank (MRR) is calculated by finding the distance of an embedded query vector to a list of possible embedded query response vectors, ordering them by cosine distance, and finding the rank of the target instance in the ordered list. The reciprocals of these ranks are summed over the testing set and then averaged by the number of testing examples. When the number of testing examples is very high, the MRR can quickly approach zero even when the rank of

Table 6: Mean Reciprocal Rank and F1 score on the testing set for models trained on Text and Speech descriptions over 5 runs. Triplet MRR is calculated from a query of the target and a positively and negatively associated test data point. Subset MRR is calculated from the target and a subset of four random test data points. The F1 score is calculated using the optimal threshold for each model.

| Model | F1 score | Triplet MRR | Subset MRR |
|---|---|---|---|
| Text 300 epochs | 0.84 | 0.85 | 0.89 |
| Transcribed Speech 300 epochs | 0.94 | 0.87 | 0.96 |
| T + TS 150 epochs | 0.92 | 0.87 | 0.94 |
| (Test on T) | - | 0.87 | 0.96 |
| (Test on TS) | - | 0.87 | 0.94 |
| wav2vec 2.0 300 epochs | 0.83 | 0.85 | 0.86 |
| MFCC + LSTM 300 epochs | 0.67 | 0.69 | 0.49 |
| Random Baseline | - | 0.61 | 0.46 |

the instance near the top of query responses, rendering the metric difficult to interpret. To counteract this and to evaluate our model on a scenario that is more realistic to what it might be used for, such as object retrieval, instead of ranking the entire testing set we rank a select few instances. Our Triplet MRR metric is calculated from a triplet of the target, positive, and negative instances and the Subset MRR is calculated from a subset of the target and four other randomly selected instances.

The combined "T + TS" model is evaluated three separate times. First, it is tested individually on held-out sets where L is drawn first from text, then from speech. It is then evaluated on the combination of the two held-out sets. From our F1 evaluation, the transcribed speech model performs better than the other models, including the text model. These results seem to indicate that, despite potential errors in the transcription process, spoken input may be more linguistically meaningful than typed input. In all testing scenarios, there is little difference between the transcribed speech model and the combined text and transcription model.

All of our models perform better on the Subset MRR task than the Triplet MRR. This is likely due to the fact that the Subset MRR task does not intentionally contain a distracting positive instance. In a realistic environment, a robot could be faced with cluttered scenes with many distracting instances, both positive and negative, that it would need to distinguish between.

We train two models for grounding speech to images using manifold alignment. The first one uses the wav2vec 2.0 [5] representations as speech features and the second one uses MFCCs. We train both models for 300 epochs. The wav2vec 2.0 model achieves comparable performance to the model trained on transcribed speech on the Triplet MRR, showcasing that the speech data in our dataset is suitable for direct grounding of speech. However, the Subset MRR results show that there is a gap in performance between the two modalities.

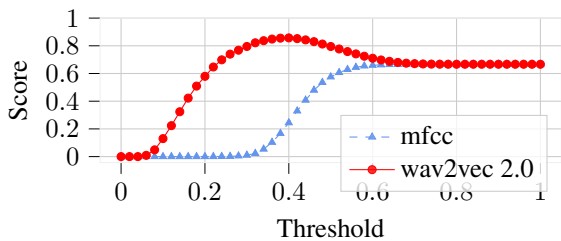

Figure 6: Threshold classification results for the speech models on the validation set. The wav2vec 2.0 model achieves a peak of 0.85 while the mfcc model stagnates at 0.66.

The MFCC model did not learn much. Figure 6 shows that the model achieves peak performance when the threshold is 1, classifying every pair as positive. The MRR results for the MFCC model in table 6 tell the same story with the model performing similarly to the random baseline. These results indicate that leveraging the semantic information learned by highly pretrained models such as wav2vec 2.0 significantly improves the quality of our grounding.

## 6    Conclusion

We introduced a new dataset that has four modalities of input (text, speech, RGB and depth) and allows us to tackle new challenges in grounded language learning such as learning directly from speech audio. Our investigation of the dataset establishes the quality of the data. Specifically we showed the feasibility of learning from typed text, transcriptions and raw speech. We also showed that the difference between the results of learning from typed or spoken descriptions is marginal. Our introduced baseline results show utility of the modalities and room for future methods to address issues not handled by current tools.

## Acknowledgments

This material is based in part upon work supported by the National Science Foundation under Grant Nos. 1637937, 1813223, 1940931, 2024878, and 1920079.

This material is also based on research that is in part supported by the Air Force Research Laboratory (AFRL), DARPA, for the KAIROS program under agreement number FA8750-19-2-1003. The U.S.Government is authorized to reproduce and distribute reprints for Governmental purposes notwithstanding any copyright notation thereon. The views and conclusions contained herein are those of the authors and should not be interpreted as necessarily representing the official policies or endorsements, either express or implied, of the Air Force Research Laboratory (AFRL), DARPA, or the U.S. Government.

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
