# OpenReview forum: "A Spoken Language Dataset of Descriptions for Speech-Based Grounded Language Learning"
_NeurIPS.cc/2021/Track/Datasets_and_Benchmarks/Round1 — NeurIPS 2021 Datasets and Benchmarks Track (Round 1)_

### Official Review · Reviewer_KsKF · 2021-07-03
**Small but carefully crafted resource for an emerging field**

**Rating:** 7
**Confidence:** 4

**Strengths:**

The dataset contains coloured 3D point clouds and RGB images of everyday objects - an input likely to be obtained from an embodied agent. Objects also are represented from four different perspectives to mitigate the effect of training based on iconic representations only. Together, this makes the dataset particularly well suited for research on learning in embodied agents.
Object descriptions are provided in three variants: text-based, in spoken language, and automatically transcribed. As natural audio descriptions are costly to collect, any dataset dedicated to this type of data is a valuable asset. The three-type setup in addition also allows for comparing training with these three different inputs on embodied language learning. This might yield interesting insights because in real-world situations, descriptions are likely to be understood imperfectly and therefore fully correct text-based descriptions might not be obtainable. With their experiments, the authors provide initial evidence that - with current ASR techniques - this might be less of an issue.

The collection of the data is described clearly, and the data itself is structured well and straightforward to process. Recording audio from crowdsourced participants always is a privacy risk, but the authors did their best to mitigate this by clearly informing participants and collecting fact-based audio descriptions only that - even if traced to a speaker - are likely too benign to cause any harm. The authors even went so far as to hold out large portions of the dataset as the consent form might have malfunctioned for a portion of the data collection.

Spoken descriptions were annotated with respect to audio and voice quality as well as dialect by the authors. While there is not too much information on this aspect in the paper, this might be an interesting avenue for future research regarding bias in ASR and the processing of spoken language in general.



**Weaknesses:**

A current limitation is the relatively small size of the dataset - something the authors are set to address by aiming to triple the number of entries for publication.

In terms of evaluation, the paper reports model performance using a mock-up retrieval task only. In order to set a proper benchmark, I would recommend coupling the dataset as a train set with an external (pre-existing) retrieval task to realise its full potential as a benchmark.

Having conducted a range of data collection studies on AMT myself, I’d like to see a little more info on the (expected) hourly pay of participants, required qualifications (e.g. region, language, education or AMT stats) and whether (and how) the authors address the almost inevitable noise in the data collected.

The section on annotating the spoken language descriptions needs extending if it is to be made a proper contribution to the work in the paper.


**Additional Feedback:**

You clearly want to highlight the different features and potential applications of your dataset. Maybe try to sharpen your focus a little more on a selection of those features to improve coherence. Otherwise it’s a great resource and a well-written paper.


**Clarity:**

The paper is clear and well written.


**Correctness:**

The dataset is carefully designed and collected in a sound way. The paper clearly reports the data’s key features and statistics. As mentioned earlier, I would recommend coupling the dataset as a train set with an external (pre-existing) retrieval task to realise its full potential as a benchmark.


**Documentation:**

The data is available online on a well-known platform. The collection of the data is described clearly, and the data itself is structured well and straightforward to process. The paper clearly reports the data’s key features and statistics.


**Ethics:**

The collection of the data is described clearly, and the data itself is structured well and straightforward to process. Recording audio from crowdsourced participants always is a privacy risk, but the authors did their best to mitigate this by clearly informing participants and collecting fact-based audio descriptions only that - even if traced to a speaker - are likely too benign to cause any harm. The authors even went so far as to hold out large portions of the dataset as the consent form might have malfunctioned for a portion of the data collection.


**Relation To Prior Work:**

I don’t have a great overview of the current state of the art or other resources in grounded language acquisition, but my impression is that the authors clearly position their work in relation to previous work and the field in general.


**Summary And Contributions:**

The paper presents a dataset of 207 everyday objects, each captured through pictures and (coloured) 3D point clouds from four different angles. Additionally, every object has been described a number of times by crowdsourced annotators, either in writing or in spoken language. As a result, the dataset lends itself particularly well for researching grounded language acquisition for embodied agents.

On a mock-up retrieval task, automatically transcribed descriptions performed on par with text-based descriptions proper, and training directly with speech input decreased performance by a few percent points. The dataset therefore could be used to compare grounded training using speech and traditional text-based approaches, but is likely to require coupling with an external task to reach its full potential as a benchmark.

The authors aim to triple the size of the dataset for publication.

---

> ### Author Response · Authors · 2021-07-11
> **Author response**
>
> We thank the reviewer for their constructive and positive remarks. We are pleased that the reviewer appreciated our efforts in navigating the ethical issues that can arise in collecting data from crowdsourced participants and found our speaker annotations to be helpful in regards to bias studies in ASR and speech processing. We are very hopeful that our dataset will allow many researchers to explore different aspects of these problems.
>
> **Re: External retrieval task:**
> >I would recommend coupling the dataset as a train set with an external (pre-existing) retrieval task to realise its full potential as a benchmark.
>
> We thank the reviewer for the external retrieval task idea. Picking a good pair of datasets and tasks may prove challenging, as classes may not overlap. Resolving those details may exceed the scope of a signal paper but we will investigate options.
>
> **Re: Extra information on AMT data collection:**
> >Having conducted a range of data collection studies on AMT myself, I’d like to see a little more info on the (expected) hourly pay of participants, required qualifications (e.g. region, language, education or AMT stats) and whether (and how) the authors address the almost inevitable noise in the data collected.
>
> Regarding the data collection on AMT, the reviewer is right about the inevitable presence of noisy data. While we were careful to not limit the participants in their descriptions, we provided them with a few rules. We asked them to not provide single-word descriptions and to focus on the object and not describe the background. Examples from users that did not follow those guidelines were filtered out. We limited the textual description collection task to participants in the United States. The workers were paid \\$0.13 per hit for text descriptions of five objects, and \\$0.08 for spoken descriptions of one object.
>
> **Re: Extending spoken analysis section:**
> > The section on annotating the spoken language descriptions needs extending if it is to be made a proper contribution to the work in the paper.
>
> Our goal in the spoken analysis section is to provide the results necessary to show that the data is valuable and that we have taken care to explore these diverse issues in the construction and curation of the data. Could the reviewer clarify what aspects of the spoken analysis they would like us to extend?

---

> > ### Comment · Reviewer_KsKF · 2021-07-20
> > **Clarification**
> >
> > My comment about "The section on annotating the spoken language descriptions..." specifically refers to section 3.2.1. I think it's useful to annotate collected speech data with respect to gender/accent/voice qualities/noise etc. - but I'd argue that they unfold their full potential when compared to the characteristics of other datasets - which arguably is difficult as long as others don't report them as well - or when these characteristics are used to split the data into different train and test sets that can showcase a model's difficulties when training on speech data without accent only and then testing it on data with accents etc.

---

### Official Review · Reviewer_829p · 2021-07-04
**A solid dataset work concerning various modalities, especially RGB, depth, digitized text and spoken language**

**Rating:** 7
**Confidence:** 4

**Strengths:**

- The relation to the prior work is well stated and why this work is necessary is well demonstrated
- The proposed dataset deals with the transcript with errors, which is far effective in real world setting compared to clean text
- The construction bases on household objects which are welcomed for practical use
- Metadata for speech files may help mitigate the possible bias regarding demographics of the speakers

**Weaknesses:**

- Though the dataset contains multimodality, the descriptive sentences does not seem to make the most of multimodal setting due to their content being simple. This does not necessarily harm the merit of this paper, but it would have been better if the spoken descriptions could contain more non-verbal information compared to the text ones.

**Additional Feedback:**

It would be better if it is stated that the provider of spoken descriptions are informed of the fact that their voice is to be annotated with their attributes.

**Clarity:**

The paper is well written, though the content is a bit difficult to follow for those who are not familiar with grounded language.

**Correctness:**

The paper incorporates substantial evaluation for the proposed benchmark, with classification and retrieval tasks, and with appropriate models.

**Documentation:**

There are sufficient supporting materials suggested for the proposed benchmark.

**Ethics:**

It seems that most concerns are resolved in the statement section. One suggestion is written in the comments.

**Relation To Prior Work:**

The related work is well organized and it seems that the proposed dataset widens and integrates the prior materials.

**Summary And Contributions:**

This paper suggests a well-constructed multimodal, multi-labelled dataset for household object description. The modalities include RBG with depth, typed language (ground truth), speech audio and its transcript (total 4). The paper describes why those modalities are chosen, with which content the dataset is created upon, the procedure and some analytic experiments. Beyond just that the proposed dataset contains various modalities, it gives a link between those modalities by grounding on the same description, and roughly shows how those modalities can contribute to the downstream tasks. Whole procedure is clear, and the dataset seems to be a nice contribution for the community that handles grounded language.

---

> ### Author Response · Authors · 2021-07-11
> **Author response**
>
> We thank the reviewer for their in-depth positive feedback. We appreciate that the reviewer acknowledges our dataset as a necessary resource for grounded language researchers. Our focus is to have a realistic dataset of household objects that can be used to train models for robots to perform in real, noisy situations. We are particularly pleased that the reviewer mentions the potential of the dataset in studying the effects of and alleviating bias in language grounding models.
>
> **Re: Non-verbal information in spoken descriptions:**
> >Though the dataset contains multimodality, the descriptive sentences does not seem to make the most of multimodal setting due to their content being simple
>
> Could the reviewer clarify what non-verbal information they are looking for in the spoken descriptions? We agree that more complex objects and scenes are desirable for future work. Our hope is that the point cloud information will allow augmentation of this data into more complex scenes in the future.
>
> **Re: Informing subjects of speech annotations:**
> >It would be better if it is stated that the provider of spoken descriptions are informed of the fact that their voice is to be annotated with their attributes.
>
> We will clarify that our IRB has approved the protocol involved in the data collection process. We do note that making subjects aware of the precise details of what is being annotated can itself cause subjects to change their behavior. Our goal was to try and get spoken responses in as natural a state as possible, given the constraints of the platforms in use. Consideration of how additional disclosure may impact the received response will be appropriate for future data collection runs.

---

> ### Comment · Reviewer_829p · 2021-07-20
> **Question resolved**
>
> The reviewer's question is resolved by author comment, and future development is expected by adding more complex scenes. Thanks for the nice work.

---

### Official Review · Reviewer_Z1Fr · 2021-07-04
**a multimodal dataset of objects with image frames, depth point cloud, speech and text**

**Rating:** 7
**Confidence:** 4
**Clarity:** This paper is well written and easy t…

**Strengths:**

1. A public available multimodal (image frames, depth point could, speech and text) dataset of household objects is proposed.
2. Experiment is conducted to analyze the difference between the speech and text descriptions and find that language groundings from transcribed speech perform similarly to models trained on text.


**Weaknesses:**

1. There are some other better speech pre-training models, such as vave2vec2.0 and hubert. I think they could be used instead of vq-wav2vec.
2. Some part of the experiment is not clear:1) I am not sure the queries for grounded language are image frames or depth point. 2) for retrieval, when few instances are selected, how many instances are selected for each query,  andhow many positive and negative samples in them?


**Additional Feedback:**

Is it possible to map inputs of different modalities into a discrete space, such as the VQ space?

**Correctness:**

I am curious about the experiment result using some types of objects for testing which are not shown in the training data. Is it much worse than the current setting or not?

**Documentation:**

The process of dataset construction is well documented.

**Ethics:**

I don't think there is an ethics problem with this dataset.

**Relation To Prior Work:**

The discussion between this paper and other multimodal datasets should be added.

**Summary And Contributions:**

This paper proposes a multimodal dataset of RGB+depth objects with speech and text descriptions. Experiments conducted on this dataset to analyze the difference between the speech and text descriptions.

---

> ### Author Response · Authors · 2021-07-11
> **Author response**
>
> We thank the reviewer for the thoughtful review and the positive marks for our dataset being public and multimodal. We sought to pick a set of initial experiments that showed the utility and validity of the data in broad terms. We envision many nuanced research directions (eg. cross-modality domain adaptation, bias in ASR, etc.) that we hope this dataset will enable.
>
> **Re: Speech pretraining models:**
> >There are some other better speech pre-training models, such as vave2vec2.0 and hubert. I think they could be used instead of vq-wav2vec.
>
> The reviewer is correct that additional sound speech featurizations will provide potential further insight. We will include wav2vec 2.0 results in the next updated version of the paper. From our current wave2vec 2.0 experiments, we find that it performs slightly better than vq-wav2vec. There is significant progress being made in the field of speech representation learning, and our goal is that our dataset will allow researchers to use these increasingly innovative models in the context of grounded language acquisition.
>
> **Re: Queries for grounded language:**
> >I am not sure the queries for grounded language are image frames or depth point.
>
> As we aimed to showcase the multimodal nature of our dataset in the experiments, we combine both the image and depth data for each object. This process is described in more detail in the “Vision” subsection of section 4.
>
> **Re: Instance selection in experiments:**
> >for retrieval, when few instances are selected, how many instances are selected for each query, andhow many positive and negative samples in them?
>
> We consider two different object retrieval settings. In the triplet setting, 3 instances are selected: the target object, a positive instance (same class as the target) and a negative instance. In the subset setting, 5 instances are selected: the target object and 4 negative instances.
>
> **Re: Performance on unseen object classes:**
> >I am curious about the experiment result using some types of objects for testing which are not shown in the training data. Is it much worse than the current setting or not?
>
>  We will explore the reviewer’s suggestion on unseen object classes and add the results to the final version of the paper.
>
> **Re: Mapping modalities into discrete space:**
> >Is it possible to map inputs of different modalities into a discrete space, such as the VQ space?
>
> The discrete space idea is interesting, and we hope our dataset will allow further exploration of this question.

---

### Note · ~Gaoussou_Youssouf_Kebe1 · 2021-10-30

https://github.com/iral-lab/gold

---

### Decision · Program_Chairs · 2021-07-26

**Decision:**

Accept

**Comment:**

Reviewers univocally agree that this paper suggests a well-constructed multimodal, multi-labeled datasets for household object description. Paper is also well-written, well-situated with prior work. The dataset is carefully crafted, has high potential for multiple domains. A few questions raised by reviewers are addressed/responded by authors.